# A Targeted Epigenetic Clock for the Prediction of Biological Age

**DOI:** 10.3390/cells11244044

**Published:** 2022-12-14

**Authors:** Noémie Gensous, Claudia Sala, Chiara Pirazzini, Francesco Ravaioli, Maddalena Milazzo, Katarzyna Malgorzata Kwiatkowska, Elena Marasco, Sara De Fanti, Cristina Giuliani, Camilla Pellegrini, Aurelia Santoro, Miriam Capri, Stefano Salvioli, Daniela Monti, Gastone Castellani, Claudio Franceschi, Maria Giulia Bacalini, Paolo Garagnani

**Affiliations:** 1Department of Internal Medicine and Clinical Immunology, CHU Bordeaux (Groupe Hospitalier Saint-André), 33077 Bordeaux, France; 2UMR/CNRS 5164, ImmunoConcEpT, CNRS, University of Bordeaux, 33076 Bordeaux, France; 3Department of Experimental, Diagnostic and Specialty Medicine, University of Bologna, 40138 Bologna, Italy; 4IRCCS Istituto Delle Scienze Neurologiche di Bologna, Via Altura 3, 40139 Bologna, Italy; 5Personal Genomics S.R.L., Via Roveggia, 43/B, 37134 Verona, Italy; 6Laboratory of Molecular Anthropology, Centre for Genome Biology, Department of Biological, Geological and Environmental Sciences, University of Bologna, 40126 Bologna, Italy; 7Interdepartmental Center, “Alma Mater Research Institute on Global Challenges and Climate Change (Alma Climate)”, University of Bologna, 40126 Bologna, Italy; 8Department of Experimental and Clinical Biomedical Sciences “Mario Serio”, University of Florence, 50139 Florence, Italy; 9Laboratory of Systems Medicine of Healthy Aging, Department of Applied Mathematics, Lobachevsky University, 603105 Nizhny Novgorod, Russia; 10Applied Biomedical Research Center (CRBA), S. Orsola-Malpighi Polyclinic, 40138 Bologna, Italy; 11Department of Laboratory Medicine, Clinical Chemistry, Karolinska Institutet, Karolinska University Hospital, 14152 Huddinge, Sweden

**Keywords:** biological age, epigenetics, DNA methylation, epigenetic clock

## Abstract

Epigenetic clocks were initially developed to track chronological age, but accumulating evidence indicates that they can also predict biological age. They are usually based on the analysis of DNA methylation by genome-wide methods, but targeted approaches, based on the assessment of a small number of CpG sites, are advisable in several settings. In this study, we developed a targeted epigenetic clock purposely optimized for the measurement of biological age. The clock includes six genomic regions mapping in *ELOVL2*, *NHLRC1*, *AIM2*, *EDARADD*, *SIRT7* and *TFAP2E* genes, selected from a re-analysis of existing microarray data, whose DNA methylation is measured by EpiTYPER assay. In healthy subjects (n = 278), epigenetic age calculated using the targeted clock was highly correlated with chronological age (Spearman correlation = 0.89). Most importantly, and in agreement with previous results from genome-wide clocks, epigenetic age was significantly higher and lower than expected in models of increased (persons with Down syndrome, n = 62) and decreased (centenarians, n = 106; centenarians’ offspring, n = 143; nutritional intervention in elderly, n = 233) biological age, respectively. These results support the potential of our targeted epigenetic clock as a new marker of biological age and open its evaluation in large cohorts to further promote the assessment of biological age in healthcare practice.

## 1. Introduction

During the last decade, DNA methylation (DNAm)-based biomarkers, designated under the term “epigenetic clocks”, have been put forward as accurate aging biomarkers. The first-generation epigenetic clocks were initially designed to predict chronological age and they have proven to be the most accurate tool to do so [1,2]. These clocks can also capture biological aspects of aging, although in some cases they exhibit only weak associations with measures of age-related decline [1,3]. Therefore, more recently, second- and third-generation DNAm-based biomarkers have been purposely developed to predict biological age and mortality risk [4,5,6,7,8]. These clocks vastly outperform the first generation predictors as markers of biological age, given their consistent association with all-cause mortality, age-related clinical phenotypes and cognitive performance measures [9,10,11,12]. A recent research has also shown that epigenetic clocks are sensitive to interventions and they can be a useful tool to screen the effectiveness of potential anti-aging drugs [13].

The epigenetic clocks described above are based on DNAm values of a large number of CpG sites across the genome, measured by Illumina Infinium microarrays. Despite their high accuracy and broad applicability to different tissues and cell types, these tools have some limitations. Their technology is characterized by a relatively high cost, both in terms of equipment and consumables, and limited accessibility. These issues could represent a constraint to their use in the context of large human cohorts and to their implementation in clinical settings. More cost-effective approaches, based on locus-targeted DNAm analysis with fewer CpG sites, have been developed [14,15]. As an example, a model built on DNA methylation values of whole blood at three CpG sites (Weidner’s estimator) was published in 2014 [15]. Based on bisulfite pyrosequencing, this model accurately predicted chronological age but not mortality in the Lothian Birth Cohort [16]. More recently, Han et al. developed targeted clocks in which DNAm was measured by droplet digital PCR or bisulfite amplicon sequencing of seven and nine target regions, respectively; these clocks showed excellent precision in the prediction of chronological age but, to the best of our knowledge, they have not been tested for age-related phenotypes indicative of biological age [14]. Large efforts toward the simplification of the epigenetic clock models have been carried out in the field of forensics, given the relatively high amount of input DNA necessary for the Illumina Infinium microarray protocol. Targeted predictive models based on quantitative PCR, pyrosequencing or Agena EpiTYPER system have been developed with promising results [17,18,19,20,21,22,23].

Here, we further add to this field by proposing a new epigenetic clock for human whole blood that has two characteristics: (1) it is based on a limited number of CpG sites, assessable by a fast and cost-effective method; (2) it reflects the duality between an accurate estimation of chronological age on one hand, and the ability to predict age-related health outcomes on the other hand. We have developed and validated such a clock, exploiting well-characterized models of increased (persons with Down syndrome) or decreased (centenarians and their offspring) biological and epigenetic age [24,25], and further evaluated its capacity to detect the impact of a one-year nutritional intervention in elderly subjects.

## 2. Materials and Methods

### 2.1. Cohorts

To identify the genomic regions to be included in our targeted assay, we used two datasets of genome-wide DNA methylation generated using the Illumina Infinium microarray (Illumina, San Diego, CA, USA; Appendix A). The first one includes whole blood samples from 29 trios composed by one person with Down syndrome (DSP), their siblings (DSS) and their mother (DSM), assessed using the Illumina Infinium450 k beadchip and publicly available in Gene Expression Omnibus (GEO) database under accession number GSE39981 [26]. The second one is an unpublished dataset including whole blood from 28 centenarians (CENT), 19 centenarians’ offspring (OFF) and 30 age-matched controls (CTR), generated using the Illumina InfiniumEPIC beadchip according to manufacturer’s instructions. Briefly, raw data were extracted using the *minfi* Bioconductor and normalized using the *preprocessFunnorm* function available in the same package [27]. To perform the EpiTYPER experiments, we used DNA from whole blood samples collected from subjects recruited in the Bologna area (Italy) in the framework of different studies and belonging to different categories (Table 1 and Table 2): (1) 315 healthy subjects from the general population, ranging from 0 to 99 years old (CTR); (2) 62 DSP ranging from 12 to 66 years old; (3) 106 CENT ranging from 100 to 112 years old; (4) 143 OFF, ranging from 55 to 89 years old. CTR, DSP, CENT and OFF samples used in the EpiTYPER experiments partially overlapped with those assessed in the Illumina Infinium experiments described above (Table 1). In addition, 124 Italian and 109 Polish old people from the intervention arm of the NU-AGE project nutritional trial (clinicaltrials.gov (accessed on 1 October 2020), NCT01754012) were included as an independent cohort to validate the clock (Table 2). This cohort was previously described [28,29]. Briefly, 1141 volunteers aged 65–79 years from five European countries (Italy, Poland, France, United Kingdom and the Netherland), free of major overt chronic diseases, were randomly assigned (1:1) to the control group (habitual diet) or intervention group (elderly-tailored Mediterranean Diet) for 1 year [30]. Baseline and after intervention biological samples and data (nutritional, clinical, health, anthropometric) were collected. Horvath’s epigenetic clock was also measured in 60 Italian and 60 Polish subjects undergoing the dietary intervention, representative of a Mediterranean and a non-Mediterranean country, respectively [31]. In the present study, we aimed at extending and confirming the previously published results.

For all the samples, genomic DNA was extracted from whole blood from venous blood samples, drawn on EDTA tubes, using the QIAamp DNA Blood Kit (Qiagen, Hilden, Germany). Five hundred nanograms of DNA were bisulfite converted using the EZ DNA Methylation Kit (Zymo Research, Irvine, CA, USA) according to manufacturer’s instructions. 

### 2.2. EpiTYPER DNAm Analysis

DNAm analysis was performed using the EpiTYPER system (Agena Bioscience, San Diego, CA, USA). Sequences of the regions of interest, flanking each selected CpG sites, were retrieved from the UCSC genome browser (https://genome.ucsc.edu/; genome assembly GRCh37/hg19, accessed on 1 September 2020). Primer design was performed using Agena Bioscience EpiDesigner software (http://epidesigner.com/; Agena Bioscience, San Diego, CA, USA; accessed on 1 October 2020), specifically optimized for the EpiTYPER system (Table 3 and Appendix A). Appendix A reports a graphical view of the CpG sites assessed by the EpiTYPER assay, including the positions of the Infinium CpG probes. Locus-targeted DNAm analysis was performed according to the manufacturer’s instructions. Ten nanograms of genomic bisulfite-converted DNA were amplified using the bisulfite-specific primers, in a 5 µL total volume using a 384-well plate. Unincorporated nucleotides and primers were then removed with the Shrimp Alkaline Phosphatase (SAP) treatment, and reverse transcription/RNaseA cleavage were performed. Finally, 20 µL of RNase-free ddH2O were added to each sample, as well as 6 mg of Clean Resin in order to eliminate salts of sodium and potassium that could interfere with the analysis. Sample dispensation on a SpectroCHIP was performed by the Nanodispenser, and final detection was conducted with the mass spectrometer. For each target region, the EpiTYPER software (Agena Bioscience, San Diego, CA, USA; software version 1.2) returns DNAm data (expressed as beta-values ranging from 0 to 1, corresponding to 0% and 100% methylated) of a number of CpG units (i.e., regions containing one or multiple CpG sites, according to the sequence of the genomic region).

### 2.3. Predictive Model and Statistical Analyses

Missing values in EpiTYPER outputs were inputted using *mice* (Multivariate Imputation by Chained Equations) R package [32]. Beta-values were converted to M-values through a logistic transformation, included in the Bioconductor package *lumi* [33]. The model to predict epigenetic age was built using a ridge regression model, included in the R package *caret* (Classification and Regression Training) [34], to regress chronological age on all the CpG units methylation levels, considering CTR from 20 to 80 years old. The predictive model was first tested using a 5-fold cross-validation procedure, dividing the cohort in training (80% of the samples) and test (20% of the samples) sets. Average Spearman correlation coefficient between predicted and chronological age was 0.89 and 0.76 in the training and test sets, respectively, while average median absolute deviation (MAD) was 4 and 5.8 in the training and test sets, respectively. We then recalculated the model using all the 278 CTR from 20 to 80 years old (Spearman correlation *p*-value = 0.89, MAD = 3.98) and applied it to the entire cohort including CTR, DSP, CENT and OFF. For each point, epigenetic age discrepancy (EAD) was calculated as the distance between epigenetic age and the regression between epigenetic age and chronological age in CTR. Positive and negative EAD values correspond to increased and decreased epigenetic ages, respectively. In the NU-AGE cohort the ridge regression model was calculated on samples at T0, then applied to the entire cohort to obtain epigenetic age values. The regression between epigenetic age and chronological age was calculated considering the samples at T0 and then used to calculate EAD values for the entire cohort. Similar results were obtained when calculating ridge regression on the entire NU-AGE cohort (Appendix A).

All the analyses were performed using R version 3.6.3 (The R Foundation for Statistical Computing, Vienna, Austria).

## 3. Results and Discussion

### 3.1. Rationale for the Selection of Target Genomic Regions

The strategy that we used to select the target regions to be included in the clock was aimed at identifying two types of candidates: (1) genomic regions whose DNAm status is highly correlated with chronological age, in order to guarantee a good correlation between predicted epigenetic age and chronological age; and (2) genomic regions whose DNAm status is correlated with chronological age but at the same time modulated in categories of subjects that, according to their clinical features or aging trajectories, have a biological age higher or lower than expected.

As a representative of the first type of genomic regions, we selected the CpG island as the promoter of *ELOVL2* gene, which encodes for the elongation of very long chain fatty acids protein 2. In 2012, we reported the strong correlation between DNAm of cg16867657 within *ELOVL2* and chronological age in whole blood [35], and since then this region has been confirmed as robustly associated with aging in several tissues [36,37]. The assessment of *ELOVL2* DNAm is largely employed in targeted epigenetic biomarkers developed for forensic applications [38,39,40,41]. Although we previously reported that hypermethylation of this genomic region in blood is associated with the prospective development of breast cancer [42], *ELOVL2* DNAm seems to depend mainly on chronological age, and Spólnicka et al. reported that no changes in *ELOVL2* DNAm occur in Alzheimer’s and Graves’ diseases [20]. 

We then searched for candidates belonging to the second category. We focused on the 353 CpG sites included in Horvath’s epigenetic clock, which were selected for their association with age using a penalized regression model [43]. We evaluated their DNAm in a dataset from persons with Down Syndrome and in a dataset from centenarians and their offspring (Appendix A). These cohorts represent well-established models of increased (Down syndrome) and decreased (long-lived individuals) biological age [44,45,46] and are therefore suitable to study the epigenetic differences associated with the discrepancy between biological and chronological age.

Down syndrome is characterized by signs of atypical aging at clinical, pathophysiological and molecular levels, and it is regarded as a segmental progeroid syndrome that mainly involves the immune and the nervous systems [44,47,48]. Several of the immunological abnormalities observed in people with Down syndrome resemble those occurring during immunosenescence and inflammation [47], and the increase in biological age is supported by different types of biomarkers measured in blood (telomere length, glycomic and epigenetic biomarkers) [25,49,50]. We and others previously demonstrated that the whole blood DNAm landscape is profoundly remodeled in Down syndrome [26,48,49,51]. Although most of these DNAm changes are distinct from those occurring during physiological age, it is likely that a certain overlap between the two conditions exists. In fact, using Horvath’s clock, we previously demonstrated that the epigenetic age of whole blood from people with Down syndrome (DSP) is higher than expected [25], an observation confirmed in subsequent studies [50,52]. Here, we considered the DNAm dataset generated by the Infinium 450 k microarray on whole blood from 29 DSP, their age-matched euploid siblings (DSS) and their mothers (DSM) [26], in which Horvath’s clock was previously assessed [25]. 

The second cohort includes centenarians (CENT) and their offspring (OFF), considered extraordinary models to study healthy aging. Centenarians have delayed morbidity, as most of them avoided or largely postponed age-related diseases [45,53]. In addition, centenarians’ offspring are healthier than people of the same birth cohort, and they have a lower risk of developing major age-related diseases and a higher probability of becoming long-lived [46,54]. Using peripheral blood mononuclear cells (PBMC), we and others previously demonstrated that long-lived individuals have lower epigenetic age than expected [24,55]. Here, we used an independent, unpublished DNAm dataset generated on whole blood (instead of PBMCs) from 28 CENT, 19 OFF and 30 controls (CTR) matched for age to OFF, using the Infinium EPIC microarray.

Among the Horvath’s 353 CpG probes, we selected those that were significantly associated with chronological age in the group of DSS and DSM and that showed significant DNAm differences in the two comparisons of DSP vs. DSS and OFF vs. CTR (Appendix A). We further refined our search by selecting only the CpG probes showing the expected sign of DNAm difference: for probes hypermethylated with age, higher DNAm in DSP respect to DSS and lower DNAm in OFF respect to CTR; conversely, for probes hypomethylated with age, lower DNAm in DSP respect to DSS and higher DNAm in OFF respect to CTR. Only 2 out 353 probes (cg13899108 and cg26372517) satisfied all these conditions. Among them, we decided to include in our targeted clock cg26372517, which maps in *TFAP2E* (Transcription Factor AP-2 Epsilon) gene, given the larger DNAm difference between the groups under investigation.

On the basis of the DNAm changes observed during aging and in the comparison of DSP vs. DSS, we further selected two additional CpG sites among the top rankers of our analysis, cg09809672 and cg22736354. The probe cg09809672 maps in the *EDARADD* (EDAR Associated Death Domain) gene and has been previously reported to be associated with aging in saliva samples [56]. *EDARADD* was also included in a panel of genes for age prediction in forensics use [57]. The probe cg22736354, which maps in *NHLRC1* (NHL Repeat Containing E3 Ubiquitin Protein Ligase 1) gene, is included in a DNAm-based forensic age predictor [58].

Finally, we decided to include in our targeted assay two additional genomic regions emerged from a deep analysis of the literature. The first region maps in the *SIRT7* (sirtuin 7) gene. Sirtuins have a crucial role in human aging and age-related diseases [59]. In particular, *SIRT7* protects from cellular senescence in human cells [60,61], and an age-related hypomethylation of *SIRT7* was described in mice livers [62]. In our datasets, we found that two adjacent CpG probes within *SIRT7* promoter (cg07855221 and cg09253473) were negatively associated with age and were further hypomethylated in DSP compared to DSS (Appendix A). We therefore included these sites, which map also in the promoter of the gene *MAFG* (MAF BZIP Transcription Factor G) in our list of targeted loci. The second region maps in *AIM2* (Absent in Melanoma 2) gene, which codes for an interferon-gamma-induced protein involved in the innate immune response. DNAm at CpG probe cg10636246 within *AIM2* was found to be associated with C-Reactive Protein serum levels [63], and it is likely to be informative of inflammation, the chronic, low-grade inflammatory status characteristic of the elderly that largely contributes to age-related diseases [64].

Notably, *ELOVL2* did not show significant changes in DSP or OFF compared to age-matched controls, confirming that its methylation is mainly associated with chronological, rather than biological, age.

In summary, we selected 7 Illumina Infinium CpG probes mapping in 6 target regions to be included in the targeted epigenetic clock (Figure 1).

### 3.2. Design of the Targeted Assay

To evaluate the methylation of the selected CpG sites, we used the EpiTYPER assay, a bisulfite sequencing method based on MALDI-TOF mass spectrometry for the quantitative and high-throughput measurement of DNAm of target genomic regions amplified by PCR. The EpiTYPER assay enables the quantification of DNAm not only of the CpG sites corresponding to the Infinium probes, but also of most of the surrounding CpG sites included in the same PCR amplicon (Table 3 and Appendix A). As the methylation of nearby CpG sites tends to be correlated [65], this approach allows *de facto* to increase the number of CpG sites potentially contributing to the predictive model, thus expanding the informativeness of the assay. 

We designed an EpiTYPER assay based on 6 PCR amplicons, including 70 CpG units (Section 2), corresponding to 121 unique CpG sites. The distribution of the 70 CpG units was as follows: 15 CpG units for *ELOVL2*, 21 CpG units for *NHLRC1*, 6 CpG units for *MAFG/SIRT7*, 7 CpG units for *AIM2*, 5 CpG units for *EDARADD* and 16 CpG units for *TFAP2E*.

### 3.3. Age Prediction Using the Targeted Epigenetic Clock

We applied the above-described EpiTYPER assay to whole blood samples from a large cohort of healthy individuals (n = 315) ranging from 0 to 98 years old (Table 1). 

For each of the 70 CpG units returned by the assay, we first evaluated Spearman correlation between DNAm values and chronological age. A significant correlation (Spearman correlation *p*-value < 0.05) was found in 48 CpG units (1 in *AIM2*, 9 in *NHLRC1*, 6 in *MAFG/SIRT7*, 15 in *ELOVL2*, 4 in *EDARADD* and 13 in *TFAP2E*). The CpG unit with the most significant correlation with age in each target region is reported in Figure 2A. 

We then investigated the feasibility of building a predictor of age using the DNA methylation values of the six selected regions. On the basis of previous studies indicating that epigenetic age changes in a linear way between 20 and 80 years old [66], we restricted our analysis to individuals in this age range (278 subjects). We then reiteratively divided the dataset in train (80% of the total number of individuals) and test (20% of the total number of individuals) subgroups, balanced for age (Section 2). In each iteration, we applied ridge regression to the train dataset and inferred epigenetic age in train and test datasets. In the training sets, the mean Spearman correlation coefficient between epigenetic and chronological ages was 0.89, while Mean Absolute Deviation (MAD) between epigenetic and chronological age was 4 years. In the test sets, the mean Spearman correlation coefficient and MAD were 0.76 and 5.8 years, respectively. MAD values tended to be slightly larger but still comparable to those of previously published untargeted and targeted epigenetic clocks [67], indicating that the DNAm of the regions that we selected can provide a reliable age prediction. 

We thus applied ridge regression to the entire dataset of 278 healthy individuals from 20 to 80 years old. The model provided an accurate estimation of chronological age (Spearman correlation coefficient = 0.89; MAD = 3.98 years) (Figure 2B).

### 3.4. Application of the Targeted Epigenetic Clock to Models of Increased and Decreased Biological Age

We then assessed whether the targeted epigenetic clock and the predictive model described in the previous paragraph could be informative of biological age. To this aim, we first considered the same categories of individuals investigated in the step of Infinium probes selection but including a larger number of individuals (Table 1). We used the targeted epigenetic clock to estimate epigenetic age in whole blood samples from 62 DSP, 106 CENT and 143 OFF, in addition to the 278 CTR evaluated above (Figure 3A). For 53 CTR, 11 DSP, 22 CENT and 53 OFF both the epigenetic ages estimated using Horvath’s clock and the epigenetic ages estimated using our targeted clock were available. The two measures were well correlated (Spearman correlation coefficient 0.81. *p*-value < 0.001; Appendix A), confirming the validity of our approach.

We calculated the epigenetic age discrepancy (EAD) for each sample as the residual of the regression between epigenetic age and chronological age calculated in control samples (Section 2, Table 1 and Figure 3B). 

In DSP, mean EAD was of 11.02 years, significantly higher than in controls (*p*-value < 0.0001, Wilcoxon-Mann–Whitney test). On the contrary, both CENT and OFF resulted significantly epigenetically younger than CTR, with mean EAD equal to −6.45 (*p*-value < 0.0001, Wilcoxon-Mann–Whitney test) and −1.65 years (*p*-value = 0.015, Wilcoxon-Mann–Whitney test), respectively (Table 1 and Figure 3B). Mean EAD values observed in DSP, CENT and OFF were comparable to those previously obtained with the Horvath’s clock in partially overlapping cohorts [24,25].

Collectively, these data show that the targeted epigenetic clock that we developed recapitulates previous studies in which Horvath’s epigenetic clock was applied to the same categories of subjects [24,25]. Although this result supports the potential of our targeted epigenetic clock as marker of biological age, it should be considered that DSP, CENT and OFF were used for the selection of the CpG sites that we included in the target assay. For this reason, we validated the targeted epigenetic clock in a completely independent cohort, as discussed in the next session.

### 3.5. Application of the Targeted Epigenetic Clock to an Independent Validation Dataset

We previously described the impact of a one-year Mediterranean-like diet on epigenetic age acceleration measures assessed with Horvath’s model [31]. Briefly, within the framework of the European project NU-AGE [28], we observed an epigenetic rejuvenation of participants (60 Italian and 60 Polish subjects) after one year of nutritional intervention [31].

We measured our new targeted epigenetic clock in whole blood samples from the same study, but increasing the number of subjects analyzed (124 Italian and 109 Polish subjects; Table 2). For each individual, we calculated the epigenetic age at baseline (T0) and after one year of nutritional intervention (T1), as well as the measures of EAD at both time points (Section 2, Table 2). 

Epigenetic age was significantly associated with chronological age at T0 (*p*-value < 0.001, Spearman correlation coefficient = 0.62, MAD = 2.2 years) (Figure 4A). We then evaluated the impact of one-year Mediterranean-like diet by comparing for each subject EAD values at T0 and T1. We found that EAD values were significantly lower at T1 with respect to T0 (Figure 4B), both when performing an unpaired (Wilcoxon-Mann–Whitney test *p*-value = 0.004; Welch’s *t*-test *p*-value = 0.016) and a paired (Student’s paired *t*-test *p*-value = 0.007) analysis. This result indicates that our targeted epigenetic clock detects a significant rejuvenation after one year of nutritional intervention. The mean of the differences between EAD values at T1 and EAD values at T0 for each subject was −0.58 years, a value comparable to the extent of rejuvenation that we previously observed using Horvath’s clock in the same cohort [31]. Collectively these results suggest that our targeted epigenetic clock is effective in detecting small changes in epigenetic age, such as those expected after a one-year nutritional intervention.

The rejuvenation effect was also evident when subdividing the cohort by country (Italy and Poland) and sex, although it reached statistical significance only in Italian and in Italian males (Student’s paired *t*-test *p*-value = 0.03; Appendix A). 

## 4. Conclusions

Targeted epigenetic clocks are cost-effective and high-throughput alternatives to algorithms based on DNAm measurement by genome-wide and whole genome approaches. Several targeted epigenetic clocks have been developed so far, selecting the genomic regions in order to maximize the accuracy of prediction of chronological age [67]. While these clocks have a remarkable relevance in the forensics field, their informativeness in the study of human aging and age-related diseases is limited, as the main aim in this case is the prediction of biological age, over chronological age. 

In the present study, we developed and applied a targeted epigenetic clock purposely optimized for the measurement of biological age. The strategy that we used to select the targets of our assay was to combine genomic regions with a high degree of correlation between DNA methylation and chronological age, together with genomic regions that could also reflect differences in aging trajectories among individuals. The resulting clock, based on the methylation of six genomic regions, showed a good accuracy in predicting chronological age in healthy individuals. Most importantly, we demonstrated that the deviation between epigenetic and chronological age was informative of the biological age of individuals in three different conditions (Down syndrome, longevity and nutritional intervention optimized for the elderly population) that were assessed with Horvath’s clock in previous studies [24,25,31]. The results of the targeted epigenetic clock were comparable to those of Horvath’s clock in terms of both direction and extent of the deviation between biological and chronological age. 

Respect to genome-wide epigenetic clocks, which include CpG probes widespread across the genome, targeted epigenetic clocks have the advantage of measuring DNAm of several CpG sites adjacent in the same genomic region, whose methylation is likely to be highly correlated. This is cost-effective, as a large number of CpG sites (in our case 70 CpG units) measured with few assays (in our case six) can contribute to the prediction model. Furthermore, it should be considered that in genome-wide clocks the CpG sites included in the model represent a small fraction of the total number of probes of the microarray. Therefore, if the aim of the analysis is not an epigenome-wide association study but exclusively the evaluation of the epigenetic age of an individual, the cost of the genome-wide approach is not justified. The targeted epigenetic clock that we developed in the present study represents, therefore, a high-throughput and cost-effective alternative for the evaluation of biological age.

We acknowledge that our study presents some limitations. The analyses that we performed do not take into account potential confounding factors such as changes in blood cell counts, as this information was not available for many of the evaluated samples. Alterations in blood cell counts can affect DNAm measurements [68] and the prediction of the epigenetic age [69]. Therefore, future studies should assess our targeted epigenetic clock using blood counts as covariates, using experimentally derived data or predicting this information by targeted DNAm assays, as recently suggested [70]. The target regions included in our clock were selected from datasets generated in whole blood, which is currently the tissue most frequently used to assess biological age. However, future studies should evaluate their performance in other accessible tissues, such as saliva, buccal swab and peripheral blood mononuclear cells (PBMC), for which at present large epigenome-wide studies on aging are not available. In addition, our clock was trained and tested in a relatively small cohort of controls and validated only in three conditions, two of which were used to develop the clock itself. The association between epigenetic age predicted by our targeted epigenetic clock and chronological age should be tested in larger cohorts. Most importantly future studies should validate the targeted epigenetic clock in other human models of increased and decreased aging (possibly already evaluated by the canonical Horvath’s epigenetic clocks) and assess its potential as a biomarker of morbidity and mortality.

To the best of our knowledge this is the first example of an epigenetic clock specifically designed to measure biological age by the assessment of DNAm of a small number of genomic regions. We used the EpiTYPER assay to measure DNAm, but other analogous experimental approaches, such as targeted next-generation bisulfite sequencing, could be used to quantify DNAm of the genomic regions included in our biomarker at single-base resolution. 

Biological age evaluation has the potential to complement chronological age in risk assessment in a broad spectrum of diseases and conditions. To reach this goal it is mandatory to investigate such a class of markers in very large cohorts of individuals. In this perspective, biological age protocols capable of reducing the cost and time for the analysis, such as the one that we presented here, could have a massive effect on the introduction of such measurements in a broad spectrum of clinical areas. In conclusion, we believe that our study could pave the way for the optimization and application of targeted epigenetic markers of biological age in large human cohorts and in drug-discovery pipelines.

## Figures and Tables

**Figure 1 cells-11-04044-f001:**
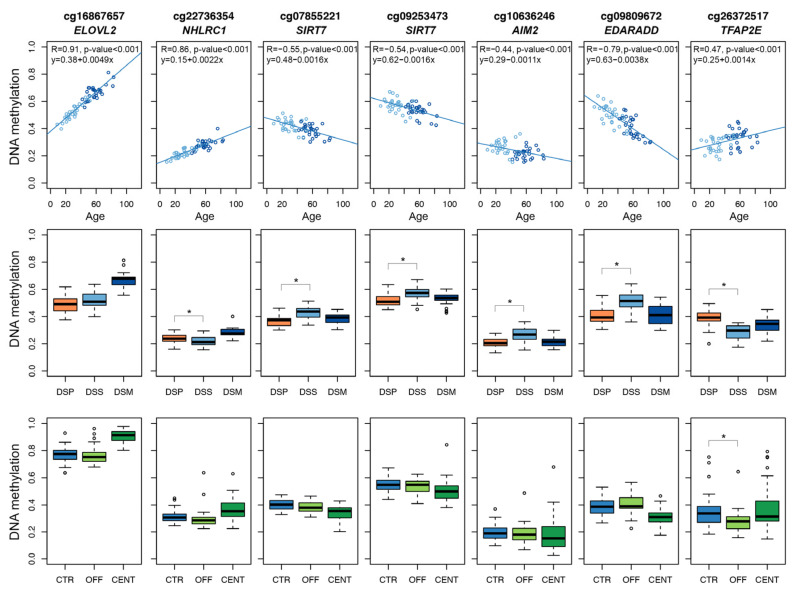
DNAm profiles of the 7 Infinium CpG probes selected for developing the targeted epigenetic assay. Each dot represents a subject. Upper panels: scatter-plots of DNAm values vs. chronological age in DSS and DSM; Spearman correlation coefficient and the equation of the regression line are reported for each CpG probe. Middle panels: boxplots of DNAm in DSP, DSS and DSM. Lower panels: boxplots of DNAm values in CENT, OFF and CTR. For the comparisons DSP vs. DSS and OFF vs. CTR, asterisks indicate statistically significant differences (*p*-value < 0.05).

**Figure 2 cells-11-04044-f002:**
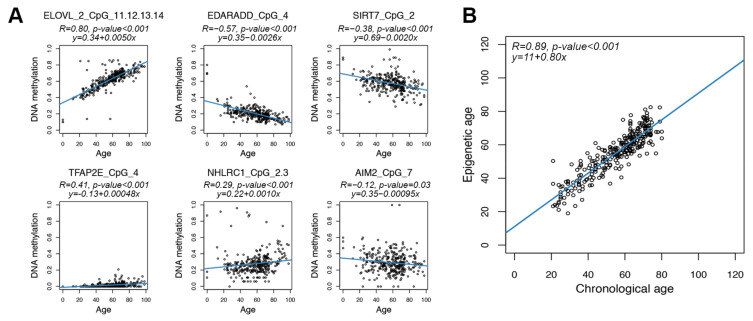
Age prediction using the targeted epigenetic clock. Each dot represents a subject. (**A**) Scatter plots of DNAm values vs. chronological age for the 6 regions assessed by the targeted assay. For each region, the CpG unit with the most significant Spearman correlation is reported. Blue lines represent linear regressions. For each CpG unit, Spearman correlation coefficient and the equation of the regression line are reported. (**B**) Epigenetic age (*y*-axis) vs. chronological age (*x*-axis) in CTR. The blue line represents linear regression. Spearman correlation coefficient and the equation of the regression line between epigenetic and chronological age are reported.

**Figure 3 cells-11-04044-f003:**
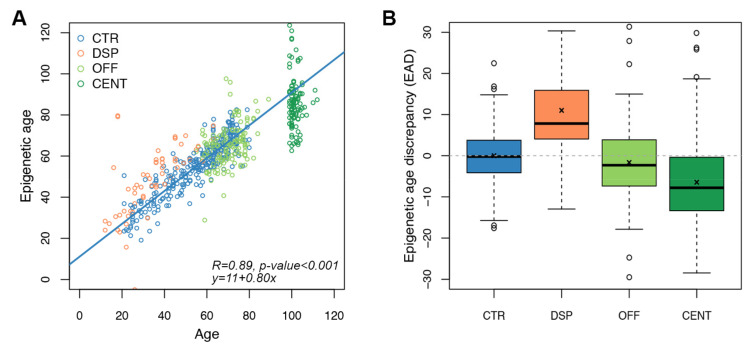
Application of the targeted epigenetic clock to increased and decreased biological age models. Each dot represents a subject. (**A**) Epigenetic age (*y*-axis) vs. chronological age (*x*-axis) in CTR, DSP (model of increased biological age), and in OFF/CENT (model of decreased biological age). Spearman correlation coefficient and the equation of the regression line are reported (**B**) Boxplots of epigenetic age discrepancy (EAD) values in the four categories of subjects considered in the validation of the targeted epigenetic clock; mean is indicated by the x symbol.

**Figure 4 cells-11-04044-f004:**
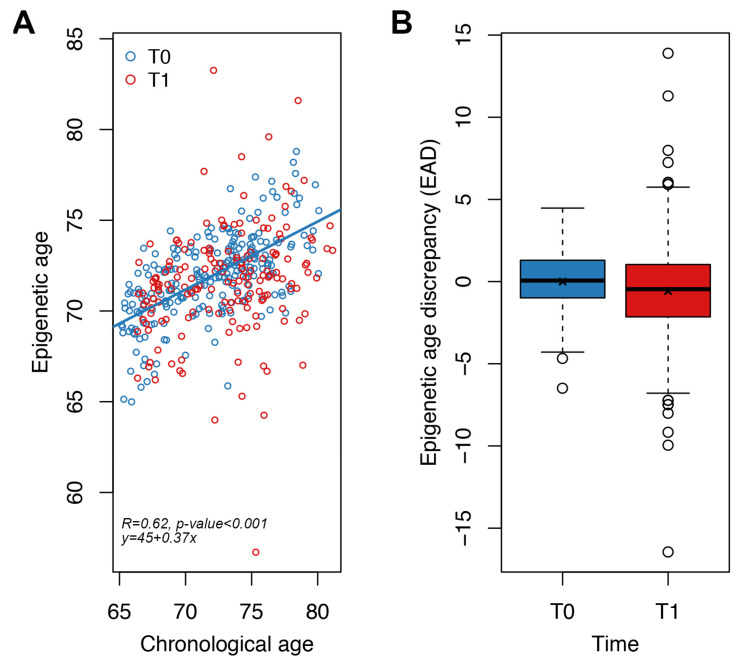
Application of the targeted epigenetic clock to an independent validation dataset. Each dot represents a subject. (**A**) Epigenetic age (*y*-axis) vs. chronological age (*x*-axis) in NU-AGE subjects at T0 and T1. Spearman correlation coefficient and the equation of the regression line are reported. (**B**) Boxplots of epigenetic age discrepancy (EAD) values at T0 and T1; mean is indicated by the x symbol.

**Table 1 cells-11-04044-t001:** Datasets used for developing and validating the targeted epigenetic clock.

Group	N (F: Females, M: Males	Overlap with Infinium Microarray Data [26]	Age Range (Years) (Mean ± SD ^1^)	Epigenetic Age (Years) (Mean ± SD ^1^)	EAD ^2^ (Years) (Mean ± SD ^1^)	*p*-Value ^3^
Controls (entire cohort)	315 (132 F, 180 M, 3 NA)	32 subjects	0–98 years (57.32 ± 18.71)	-	-	-
Controls (age range 20–80 years)	278 (117 F, 161 M)	32 subjects	21–80 years (54.96 ± 15.03)	54.96 ± 13.43	0 ± 6.04	-
Persons with Down syndrome	62 (27 F, 35 M)	11 subjects	12–66 years (33.97 ± 13.46)	49.23 ± 34.94	+11.02 ± 33.33	<0.001
Centenarians	106 (82 F, 24 M)	11 subjects	100–112 years (101.5 ± 2.44)	85.66 ± 12.23	−6.45 ± 12.43	<0.001
Centenarians’ offspring	143 (81 F, 62 M)	19 subjects	55–89 years (70.06 ± 6.69)	65.35 ± 9.75	−1.65 ± 8.96	0.015

^1^ SD: standard deviation. ^2^ EAD: Epigenetic age discrepancy. ^3^
*p*-value resulting from the comparison of EAD values between each category of subjects and control subjects ranging from 20 to 80 years (Wilcoxon-Mann–Whitney test).

**Table 2 cells-11-04044-t002:** Dataset used for the independent validation of the targeted epigenetic clock.

	All	Italy	Poland
Subjects (N)	233	124	109
Overlap with Infinium microarray data [31]	120	60	60
Males/Females (N)	105/128	60/64	45/64
Chronological age at T0 (years) mean ± SD ^1^	71.89 ± 3.91	72.16 ± 3.79	71.58 ± 4.03
Epigenetic age at T0 (years) mean ± SD ^1^	71.89 ± 2.34	71.84 ± 2.19	71.94 ± 2.51
EAD ^2^ at T0 (years) mean ± SD ^1^	0.00 ± 1.82	−0.14 ± 1.69	0.17 ± 1.96
Chronological age at T1 (years) mean ± SD ^1^	72.93 ± 3.91	73.22 ± 3.78	72.59 ± 4.03
Epigenetic age at T1 (years) mean ± SD ^1^	71.70 ± 3.11	71.51 ± 2.97	71.92 ± 3.26
EAD ^2^ at T1 (years) mean ± SD ^1^	−0.58 ± 3.16	−0.87 ± 3.03	−0.23 ± 3.28

^1^ SD: standard deviation. ^2^ EAD: Epigenetic age discrepancy.

**Table 3 cells-11-04044-t003:** CpG probes and genomic regions assessed in the targeted epigenetic clock.

CpG Probe	Location	Associated Gene	Region Assessed in the Targeted Assay	Assessable CpG Units
cg16867657	chr6:11,044,877	*ELOVL2*	chr6:11,044,680–11,045,053	15
cg22736354	chr6:18,122,719	*NHLRC1*	chr6:18,122,552–18,123,149	21
cg07855221	chr17:79,877,314	*SIRT7/MAFG*	chr17:79,877,158–79,877,497	6
cg09253473	chr17:79,877,390	*SIRT7/MAFG*	chr17:79,877,158–79,877,497	6
cg10636246	chr1:159,046,973	*AIM2*	chr1:159,046,884–159,047,270	7
cg09809672	chr1:236,557,683	*EDARADD*	chr1:236,557,384–236,557,805	5
cg26372517	chr1:36,039,159	*TFAP2E*	chr1:36,038,876–36,039,325	16

## Data Availability

Illumina Infinium dataset on DNA methylation in persons with Down syndrome is available in Gene Expression Omnibus (GEO) database under accession number GSE39981. All the other data are available upon reasonable request.

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
