# Peer review of "A Targeted Epigenetic Clock for the Prediction of Biological Age"

_cells, 2022, doi:10.3390/cells11244044_

Round 1

Reviewer 1 Report

I can identify three major issues with the manuscript as is:

1) using the Down syndrome as an accelerated ageing condition does not seem appropriate since even the authors note that it is only a partial progeroid syndrome. In addition, the life expectancy of people with Down syndrome is (2007) half of that of centenarians, yet the "epigenetic age" of both categories overlap Fig. 3A. Finally, lines 190-191, the use of unsuccessful aging” to describe aging with the Down syndrome sounds rather offensive.

2) the authors used data obtained in a previous clinical trial (NCT01754012) to validate their clock. Is it ethically valid?

3) they have used an "hypothesis-driven strategy" to choose their regions of interest but a comparison with randomly selected regions would have been greatly appreciated in order to estimate the gain of this approach.

4) xy plots and linear regression miss correlation values (and equations?). It would be of importance, especially for Fig2A where the correlations seem quite poor.

5) The effect of a 1 year Mediterranean diet on the presented epigenetic clock is highly oversold and the text is not in agreement with the content of figure 4. At best, the figures show an increased dispersion in the data post treatment. It is likely that the significance of the Student’s paired t-test simply reflects this associated with the relatively large n.

6) the strong overlap between train and test datasets can let us fear a model that is strongly over fitted to the present case. The results presented figure 2A seem to go in that direction.

Overall, the manuscript is well written and easy to read but the results presented are highly overstated. The most important claim of the paper, the use of targeted vs genome-wide epigenetic clock, is barely discussed. A thorough comparison of the epigenetic clock presented here with previously described others should be the main result.

Author Response

Reviewer 1

1) using the Down syndrome as an accelerated ageing condition does not seem appropriate since even the authors note that it is only a partial progeroid syndrome. In addition, the life expectancy of people with Down syndrome is (2007) half of that of centenarians, yet the "epigenetic age" of both categories overlap Fig. 3A. Finally, lines 190-191, the use of unsuccessful aging” to describe aging with the Down syndrome sounds rather offensive.

We thank the Reviewer for this comment, which prompted us to better argument why we included Down syndrome as a model of increased biological age in our study. Indeed, Down syndrome is widely recognized as a segmental progeroid condition, in which the most affected systems are the nervous and the immune ones. Several of the immunological abnormalities observed in persons with Down syndrome resemble those occurring during immunosenescence and inflammaging (doi: 10.1007/s00281-020-00804-1). The increased biological age of blood from persons with Down syndrome is supported by the use of biomarkers, including the Horvath’s clock (doi: 10.1111/acel.12325; doi: 10.1111/acel.13652) but also not-epigenetic ones, like those based on glycomics (doi: 10.1021/acs.jproteome.5b00356) or on the measurement of telomere length (doi: 10.1089/dna.2014.2746).

The observed overlap in epigenetic age between some of the persons with Down syndrome and some of the centenarians (Figure 3A) is due to the fact that the former tend to have an increased biological age, while the latter tend to have a decreased biological age (Figure 3B).

We didn’t mean to be offensive with the term “unsuccessful aging” and we apologize for this. We replaced this expression across the text.

The first section of Results has been therefore modified according to the Reviewer’s suggestions.

2) the authors used data obtained in a previous clinical trial (NCT01754012) to validate their clock. Is it ethically valid?

The assessment of epigenetic clocks and, more in general, the analysis of DNAm profiles were foreseen in NU-AGE study (see https://clinicaltrials.gov/ct2/show/NCT01754012 and https://cordis.europa.eu/project/id/266486/reporting/it)

3) they have used an "hypothesis-driven strategy" to choose their regions of interest but a comparison with randomly selected regions would have been greatly appreciated in order to estimate the gain of this approach.

We compared the epigenetic age values calculated using our targeted epigenetic clock with those calculated using Horvath’s clock, in which the CpG sites included in the model were selected from genome-wide data on the basis of their association with age (elastic net regression), without any a priori strategy. To this aim, we compared the samples for which both the measures were available (53 CTR, 11 DSP, 22 CENT and 53 OFF). We found a good correlation between the epigenetic ages estimated using our targeted epigenetic clock and Horvath’s clock (Spearman correlation coefficient: 0.81, p-value<0.01; Supplementary Figure 3).

Furthermore, we better discussed in the Conclusions section the advantages of our approach.

4) xy plots and linear regression miss correlation values (and equations?). It would be of importance, especially for Fig2A where the correlations seem quite poor.

We added regression lines, Spearman correlation coefficients and equations to the scatter plots.

5) The effect of a 1 year Mediterranean diet on the presented epigenetic clock is highly oversold and the text is not in agreement with the content of figure 4. At best, the figures show an increased dispersion in the data post treatment. It is likely that the significance of the Student’s paired t-test simply reflects this associated with the relatively large n.

We thank the Reviewer for these comments, which prompted us to integrate Section 3.5 with additional information that support our results.

For each subject, we calculated the difference between EAD at T1 and EAD at T0. Mean difference in EAD values was -0.58 years. This value is comparable to the values that we previously obtained applying Horvath’s clock to a smaller cohort of Italian and Polish subjects from NU-AGE project (doi: 10.1007/s11357-019-00149-0, see Supplementary File 4 of the original paper). This indicates that our targeted epigenetic clock is effective in detecting small changes in epigenetic age like those expected after a nutritional intervention of a short period like one year. To better visualize the decrease in EAD values, we added the value of the mean in the boxplots of Figure 4B.

Regarding the higher dispersion at T1, this is due to the fact that, to be consistent with the procedure used in the previous paragraphs, we calculated the ridge regression model on samples at T0, then applied it to the entire cohort to obtain epigenetic age values. We tried to calculate the ridge regression model on the entire cohort, and still we observed a significant decrease in EAD at T0 (paired t-test p-value = 0.005; Supplementary Figure 2).

We also applied the unpaired t-test to our data, and still found a significant difference in EAD values at T1 and T0 (p-value = 0.016). The t-test implemented in R software environment is a Welch t-test, which does not assume that the two populations have the same variance, as it happens for EAD values at T0 and T1 calculated using our procedure. Finally, although it is true that p-values tend to be smaller with a larger sample size, it should be taken into consideration that in our case we observed a significant p-value associated with a not negligible change in EAD values, as discussed above.

The text was updated according the above considerations in Materials and Methods and in section 3.5.

6) the strong overlap between train and test datasets can let us fear a model that is strongly over fitted to the present case. The results presented figure 2A seem to go in that direction.
We discussed better in the Conclusions section that our approach should be validated in larger cohorts. We also discussed better that our aim of was not to develop a predictor of chronological age (for which over-fitting could be a problem), as many clocks of this type were developed in the forensics field. On the contrary, our aim was to develop a predictor of biological age, which could grasp the differences in aging trajectories among individuals.

Overall, the manuscript is well written and easy to read but the results presented are highly overstated. The most important claim of the paper, the use of targeted vs genome-wide epigenetic clock, is barely discussed. A thorough comparison of the epigenetic clock presented here with previously described others should be the main result.

We updated the Conclusions section according to Reviewer’s suggestions.

Reviewer 2 Report

-It’s well designed and well written article, methodology section is clear and results were laid out with the updated references from previous studies. The screened DNAm markers will be useful in delineating the differences in chronological and biological age on larger cohort with reduction of cost and resources.

-The author predicts the biological age markers by re-analysis of previous existed genome wide datasets of DNA methylation, generated from illumina Infinium microarray and then applied the screened markers on large population dataset using  Epi-typer assay on healthy subjects, Centenarians and Centenarian offspring and on patients with down syndrome. Down syndrome is characterized by signs of atypical aging at clinical, pathophysiological and molecular levels, and it is regarded as a segmental progeroid syndrome. 

The approach to screen the target for the prediction of biological age from down syndrome patients may not be the ideal dataset, as the disruption of nuclear envelop, topological and lamin associated domains (TAD and LAD) alters the acetylation, methylation pattern in a way that does not necessarily resembles with natural ageing process. I think this could be addressed in the discussion section. Although the selected screened markers would able to show significance difference when applied to a group that were intervened with Mediterranean diet, yet some group was non-significant when data was further sub-divided by country and sex. 

-The sub-division of data of figure 4, for sex and for country can be added as supplementary figure.

-Abbreviation for Down syndrome patient (DSP) is not consistent in the manuscript it sometime only abbreviated as DS (line 90, line 212, line 235, line 311 supplementary table 1 or supplementary file 1

-The epigenetic and biological age is presented as Mean and S.D? please mention that in Table 1 or in its legend.

-Two different CpG probes for sirt7 which should target the same CpG island, is mentioned in Table 2. The result section (Figure 2) is illustrated using which probe? Or the methylation was showed as the average of both probes? If there are any differences, please mention which probe is better over other

-NU-Age nutritional trial was carried on five different EU countries including (France, UK, Netherland, Poland and Italy). Verification of Biological clock was assessed on two countries though. The selection was random? Any rational for selecting Poland and Italy group?

-why DNA extracted from whole blood cells would be advantageous over DNA extracted only from PBMCs (please add one or two line discussion section)

Author Response

Reviewer 2

-It’s well designed and well written article, methodology section is clear and results were laid out with the updated references from previous studies. The screened DNAm markers will be useful in delineating the differences in chronological and biological age on larger cohort with reduction of cost and resources.

-The author predicts the biological age markers by re-analysis of previous existed genome wide datasets of DNA methylation, generated from illumina Infinium microarray and then applied the screened markers on large population dataset using  Epi-typer assay on healthy subjects, Centenarians and Centenarian offspring and on patients with down syndrome. Down syndrome is characterized by signs of atypical aging at clinical, pathophysiological and molecular levels, and it is regarded as a segmental progeroid syndrome. 

The approach to screen the target for the prediction of biological age from down syndrome patients may not be the ideal dataset, as the disruption of nuclear envelop, topological and lamin associated domains (TAD and LAD) alters the acetylation, methylation pattern in a way that does not necessarily resembles with natural ageing process. I think this could be addressed in the discussion section.

We thank the Reviewer for this observation, which prompted us to better explain in section 3.1 (the one in which we presented the Down syndrome condition) why we used it a model of increased biological and epigenetic age.

Although the selected screened markers would able to show significance difference when applied to a group that were intervened with Mediterranean diet, yet some group was non-significant when data was further sub-divided by country and sex. 

-The sub-division of data of figure 4, for sex and for country can be added as supplementary figure.

We added Supplemenatry Figure 4.

-Abbreviation for Down syndrome patient (DSP) is not consistent in the manuscript it sometime only abbreviated as DS (line 90, line 212, line 235, line 311 supplementary table 1 or supplementary file 1

We checked for this inconsistency across the text.

-The epigenetic and biological age is presented as Mean and S.D? please mention that in Table 1 or in its legend.

Table 1 has been updated according to the Reviewer’s indications.

-Two different CpG probes for sirt7 which should target the same CpG island, is mentioned in Table 2. The result section (Figure 2) is illustrated using which probe? Or the methylation was showed as the average of both probes? If there are any differences, please mention which probe is better over other

Both the probes are included in the genomic region amplified by PCR and whose methylation is measured by EpiTYPER. We added a Supplementary Figure to better explain the design of EpiTYPER experiments respect to the position of Infinium probes.

-NU-Age nutritional trial was carried on five different EU countries including (France, UK, Netherland, Poland and Italy). Verification of Biological clock was assessed on two countries though. The selection was random? Any rational for selecting Poland and Italy group?

In the original design of NU-AGE study, epigenome-wide analyses and Horvath’s clock were performed on a small subset of Italian and Polish subjects, as they were representative of a Mediterranean and a non-Mediterranean country respectively. In the present study we aimed at extending and confirming the previously published results. We updated Materials and Methods section in order to clarify this aspect.

-why DNA extracted from whole blood cells would be advantageous over DNA extracted only from PBMCs (please add one or two line discussion section)

The discussion has been updated according to the Reviewer’s useful suggestion.